**Data Availability Statement:** All data supporting the findings of this study are available within the article.

**Funding:** This study was supported by National Natural Science Foundation of China (81872005) to

# Hsa-miR-22-3p inhibits liver cancer cell EMT and cell migration/ invasion by indirectly regulating SPRY2

**Shuaishuai Cui**[1☯], **Yuanyuan Chen**[1☯], **Yunfei Guo**[1], **Xing Wang**[2], **Dahu Chen**📝[1]*

**1** School of Life Sciences and Medicine, Shandong University of Technology, Zibo, China, **2** School of Life Sciences, Jiangsu Normal University, Xuzhou, China

☯ These authors contributed equally to this work.

* dahuchen@sdut.edu.cn

## Abstract

The general mechanism for microRNAs to play biological function is through their inhibition on the expression of their target genes. In cancer, microRNAs may accelerate cell senescence, block angiogenesis, decrease energy supplies, repress tumor cell cycle and promote apoptosis to function as the tumor repressors. On the other hand, microRNAs can modulate tumor suppressor molecules to activate oncogene relevant signaling pathway to initiate tumorigenesis and promote tumor progression. By targeting different genes, miR-22 can function as either a tumor suppressor or a tumor promoter in different types of cancer. In liver cancer, miR-22 mainly functions as a tumor suppressor via its regulation on different genes. In this study, we demonstrated that miR-22 indirectly regulates SPRY2 by inhibiting CBL, an E3 ligase for SPRY2 that has been confirmed. As one of the modulators of the MAPK (mitogen-activated protein kinase)/ERK (extracellular signal-regulated kinase) signaling pathway, SPRY2 plays important roles in many developmental and physiological processes, and its deregulation has been reported in different types of cancer and shown to affect cancer development, progression, and metastasis. By inhibiting the expression of CBL, which stabilizes SPRY2, miR-22 indirectly upregulates SPRY2, thereby suppressing the epithelial-mesenchymal transition (EMT), cell migration, and invasion and decreasing the expression of liver cancer stem cell (CSC) marker genes. The inhibitory effects of miR-22 on EMT, cell migration, and invasion can be blocked by the knockdown of SPRY2 expression in miR-22 overexpressing cells. Additionally, we demonstrated that miR-22 expression inhibits the ERK signaling pathway and that this effect is due to its upregulation of SPRY2. Overall, our study revealed a novel miR-22-3p/CBL/SPRY2/ERK axis that plays an important role in EMT, cell migration, and invasion of liver cancer cells.

## Introduction

The major type of primary tumor in the liver is hepatocellular carcinoma (HCC), which accounts for about 90% of primary liver cancer and is one of the leading causes of cancer-

DC. The funders had no role in study design, data collection and analysis, decision to publish, or preparation of the manuscript.

**Competing interests:** The authors have declared that no competing interests exist.

relevant deaths worldwide [1]. HCC is the third most common cause of cancer deaths in men and the seventh most common cause in women [2]. Chronic hepatitis B virus infection, hepatitis C virus infection, and alcohol intake are strongly associated with HCC development [3]. The main choice for HCC treatment is surgical resection or liver transplantation in some deliberately selected cases. However, these therapies are not very effective in improving patient survival. Therefore, we are facing an urgent challenge to understand the molecular mechanisms of HCC for developing new therapies.

MicroRNAs (miRNAs) are small noncoding RNAs that often contain about 17–25 nucleotides. They are evolutionarily conserved and widely expressed in mammalian cells. By binding to the 3'-untranslated region (3'-UTR) of target mRNAs, miRNA triggers either cleavage or translation inhibition of their target mRNAs [4]. More and more evidence indicates that miRNAs have an important role in the regulation of different biological processes, including cancer development [5]. The deregulation or dysfunction of miRNAs often leads to cancer development. The expression of some miRNAs has been found to be related to cancer prognosis, and the possibility of these miRNAs as potential therapeutic targets is currently being studied [6,7]. Many miRNAs can function as either tumor suppressors or oncogenes, depending on their tissue-specific expression patterns and mRNA targets [8].

miR-22 is a broadly studied microRNA which has two mature forms: miR-22-5p (or miR-22*) and miR-22-3p (or miR-22). When pre-miR-22 is cleaved by Dicer, it generates 22nt miR-22-5p: miR-22-3p duplex. miR-22-3p is a functional guide strand and is complementary to the target. The other strand, miR-22-5p has been considered as a passenger strand and usually will be degraded. Recent studies found that both miR-22-5p and miR-22-3p are the functional molecules and they target on different mRNAs.

It has been found that miR-22-5p is linked to cardiovascular diseases development [9] and the self-renewal of stem cells [10]. Regarding to its role in cancer, Wu etc. found that miR-22-5p plays a tumor promoter role in lung adenocarcinoma [11]. However, in non-small-cell lung cancer, miR-22-5p has the opposite effects [12]. We barely see the reports on the role of miR-22-5p in other kinds of cancers.

In terms of the function of miR-22-3p in cancers, most of studies shows that it acts as a tumor repressor. For example, miR-22-3p targets PLAGL2 [13] or PGC1$\beta$ [14] to suppress breast cancer cell migration, invasion and tumorigenesis. Through targeting different genes, miR-22-3p plays a tumor repressor role in bladder cancer [15], gastric cancer [16], colorectal cancer [17], kidney cancer [18], and lung cancer [19].

To date, miR-22-3p has been found to have an anti-tumor role in HCC [20,21]. miR-22-3p targets *sirt1* to sustain senescence-like phenotypes of HCC cells. In vitro, overexpression of miR-22-3p dramatically inhibited the proliferation, colony formation, migration, and EMT of HepG2 cells, while knockdown of miR-22-3p had the opposite effect [22]. By targeting different genes in HCC, miR-22-3p has established various roles in HCC development. It inhibited HCC cells' migration and invasion by targeting CD147 [23] and repressed HCC metastasis by inhibiting YWHAZ [24]. It also regulated the expression of HDAC4 [25] and Galectin-1 in HCC [26]. Moreover, the expression of miR-22 was downregulated in HCC patients [25]. Serum miR-22 may serve as a novel diagnostic marker in HCC [20]. Although there are numerous reports on the function of miR-22 in HCC, additional research into the mechanisms of miR-22 in HCC is still required.

The MAPK pathway is comprised of three main kinases, MAPK kinase kinase, MAPK kinase and MAPK, which activate and phosphorylate downstream proteins. MAPK cascades are critical signaling pathways that regulate a wide variety of cellular processes, including proliferation, apoptosis, differentiation and stress responses. The extracellular signal-regulated kinases ERK1 and ERK2 are ubiquitous serine-threonine kinases that regulate cellular

signaling under both normal and pathological conditions. Extracellular signal-regulated kinase 1/2 (ERK) belongs to the mitogen-activated protein kinase (MAPK) family and is responsible for transmitting extracellular signals to intracellular targets. MAPK cascades are central signaling elements that regulate the basic cellular processes. MAPK/ERK signaling pathway can be modulated by many molecules. SPRY (Sprouty homolog) family participates in modulating MAPK/ERK signaling and contains four members. SPRY2, one of the SPRY members, has a high level of evolutionary conservation. Unlike the other three isoforms that display organ-/tissue-specific expression patterns, SPRY2 is ubiquitously expressed in embryonic and adult tissues [27]. SPRY2 contains three major domains in its structure: the Casitas B-lineage lymphoma (c-Cbl) binding domain (CBD) with a key tyrosine residue localized at its N-terminal; the serine-rich motif (SRM); and the C-terminal cysteine-rich domain (CRD) [28]. These motifs are indispensable for SPRY2 to establish its biological functions. Through selectively inhibiting ERK activation induced by growth factor, SPRY2 modulates MAPK/ERK signaling pathway and participates in regulating many developmental and physiological processes. SPRY2 deregulation has been found in various cancer types and exhibited to affect tumor development, progression, and metastasis. In HCC, the expression of SPRY2 has been reported to be consistently downregulated [29,30]. In vitro studies found that overexpression of SPRY2 inhibited HCC cell growth [31]. Inactivation of SPRY2 by overexpressing a dominant negative form of SPRY2 in the mouse liver accelerated hepatocarcinogenesis [32]. Clinical studies indicated that HCC patients without SPRY2 often had poorer survival and elevated recurrence [33].

Apart from being regulated at transcriptional and translational levels, SPRY2 is also subjected to being regulated at the post-translational level. Ubiquitination is one of the post-translational modifications. During ubiquitination, the small ubiquitin molecules are added to target proteins by a series of enzymes to specifically modify target proteins. These enzymes include ubiquitin activating enzyme E1, ubiquitin binding enzyme E2 and ubiquitin ligase E3. E3 ligase is responsible for substrate specific recognition and ubiquitin labeling and is a key regulatory enzyme in the process of ubiquitination. It has been found that SPRY2 can be modified by ubiquitination. So far, three E3 ligases have been identified for SPRY2, including c-Cbl [34], Siah2 [35] and Nedd4 [36]. Among them, CBL has been identified and proved as a target of miR-22-3p [37–39]. Although there is one miR-22-5p binding site in CBL UTR sequence, no experiment data indicates that CBL is a real target of miR-22-5p. As an important regulator for many signaling pathways, CBL also participates in regulating MAPK signaling by modifying the signal molecules, such as EGFR (Epidermal Growth Factor Receptor) [40]. CBL is involved in regulating cell function and development. It is thought that the function of CBL is largely based on its ubiquitin ligase activities that catalyze many signaling molecules for degradation. SPRY2 has been found to be a substrate of CBL [34] and participates in inhibiting EMT, cell migration and invasion [41–43].

Because CBL is a direct target of miR-22-3p and also an E3 ligase of SPRY2, we hypothesize that miR-22-3p may regulate SPRY2 by inhibiting CBL to establish its functional role in HCC development. This study aimed to investigate the role of the miR-22-3p/CBL/SPRY2 signal axis in the development of HCC.

## Materials and methods

### Cell culture

HEK293T cells (CRL-3216) and HepG2 cells (HB-8065) were supplied by ATCC, USA. HuH-7 (CL-0120) cells was purchased from, Procell, China. They were cultured in high glucose DMEM (C11995500BT, Gibco, Beijing, China) medium supplemented with 10% fetal calf

serum, 100 U/mL antibiotics including penicillin and streptomycin and housed in a humidified 5% $CO_2$ incubator at 37˚C.

## Cell viability assay

Cell viability was determined by using the LDH Cytotoxicity Detection Kit (Takara, MK401), according to the manufacturer's protocol.

## Plasmids and short hairpin RNA

The miR-22-3p expression plasmid (Cat No: HmiR0267-MR04), miRNA scrambled control clone (CmiR001-MR04-B), miR-22-3p inhibitor plasmid (HmiR-AN0332-AM03), and miRNA inhibitor scrambled control clone (CmiR-AN0001-AM03-B) were purchased from GeneCopoeia. The CBL ORF sequence was cloned into the pBABE-puro vector using the following primers:

F: 5'- ATGCGGATCCATGGCCGGCAACGTGAAGAAG -3';

R:5'-ATGCGTCGACCTATGGTGCTACATGGGCAGG -3'.

SPRY2 knockdown plasmids (HSH090285-LVRH1H) and scrambled control plasmids (CSHCTR001-LVRH1H) were obtained from GeneCopoeia. The HA-ubiquitin expression plasmid was obtained from Addgene (Plasmid#18712).

## RNA isolation and real-time quantitative PCR

Trizol reagent (Invitrogen) was used to isolate total RNA from cultured cells. miRNA was extracted according to the manufacturer's instructions using the miRVANA Kit (Ambion, Carlsbad, MA). miR-22-3p quantitative PCR (qPCR) was performed using the MystiCq® MicroRNA qPCR Assay Primer (MIRAP00049); Sigma, St. Louis, MO, USA). Real-time PCR and data collection were performed on a CFX96 instrument (Bio-Rad) according to the manufacturer's instructions. The primer sequences for qPCR analysis are as follows:

EpCAM: F: 5'- TGATCCTGACTGCGATGAGAG -3';

R: 5'- CTTGTCTGTTCTTCTGACCCC -3'.

CBL: F: 5'- TGGTGCGGTTGTGTCAGAAC -3';

R: 5'- GGTAGGTATCTGGTAGCAGGTC -3'.

SPRY2: F: 5'- ATGGCATAATCCGGGTGCAA -3';

R: 5'- TGTCGCAGATCCAGTCTGATG -3'.

18sRNA: F: 5'- CAGCCACCCGAGATTGAGCA -3'

R: 5'- TAGTAGCGACGGGCGGGTGT -3'

U6: F: 5'- GCTTCGGCAGCACATATACTAAAAT -3'

R: 5'- CGCTTCACGAATTTGCGTGTCAT -3'

The comparative Ct (2-ddCt) method was applied to calculate the relative mRNA quantity between samples. 18sRNA or U6 RNA was used as an internal control gene to normalize the amount of RNA or miRNA added to the first-strand cDNA synthesis reactions. The difference between the threshold cycle (Ct) of the target gene and the Ct of the reference gene (18sRNA or U6) of the same sample was calculated as dCt. The difference of dCt between treated cells and control cells was calculated as ddCt. The final quantitation result is presented as the fold change of target gene expression in treated cells relative to control cells, normalized to 18sRNA or U6 RNA. The expression (*E*) of each target mRNA in each sample, relative to 18S rRNA, was calculated based on the cycle threshold (Ct) using the formula $E = 2e-\Delta C$ in which $\Delta C = Ct_{target}-Ct_{18S}$ or $Ct_{U6}$.

## Migration and invasion assays

Transwell migration and Matrigel invasion assays were performed, as previously described [44]. The transwell system (Corning, New York, USA) was used for assays. Briefly, the cells ($5 \times 10^4$ cells for migration assay, $1.25 \times 10^5$ cells for invasion assay) supplemented with the serum-free medium were seeded into the upper chambers coated with (for invasion) or without (for migration) Matrigel. The lower chambers contained 500 μL of culture media plus 10% FBS. After incubation for 48 h, cells on the upper chamber side of the membrane were scraped off with cotton swabs. The migrated or invaded cells were fixed with 70% ethanol and stained with 0.5% crystal violet. The cells on the lower side of the membrane were observed under a microscope (BDS400, CNOPTEC, China), and at least five pictures were taken from each membrane randomly. The stained cells were manually counted from the pictures.

## Immunoblotting

The cells were harvested and washed with cold PBS, and then were scraped off and transferred to Eppendorf tubes. After centrifugation, the cell pellets were lysed on ice with RIPA buffer to extract total cell protein. The protein concentration was measured by BCA method. Thirty milligrams of proteins were separated by SDS-PAGE and blotted onto a nitrocellulose membrane. The following antibodies were used: anti-E-cadherin (1:1000, SC-8426; Santa Cruz Biotechnology), anti-N-cadherin (1:750, SC-271386; Santa Cruz Biotechnology), anti-SPRY2 antibody (1:1000, 14954; Cell Signaling), anti-β-actin antibody (1:5000, 610182; Sigma), anti-GAPDH antibody (1:3000, SC-47724; Santa Cruz Biotechnology), anti-p-ERK1/2 (1:750, SC-81492; Santa Cruz Biotechnology), and anti-ERK1/2 (1:1000, SC-514302; Santa Cruz Biotechnology).

## Ubiquitination assay

The HepG2 cells were grown in 10 cm dishes to semiconfluency and then transfected with HA-ubiquitin expression plasmid using X-tremeGene9 (Roche), according to the manufacturer's instructions. After 48 h of transfection, the cells were treated with MG132 (10 μM, MCE) for 6 h before collection. The collected cells were lysed, and the lysate was immunoprecipitated for 2 h with an anti-SPRY2 antibody and then with protein G-Sepharose beads (GE Healthcare, Chicago, USA) overnight. The proteins were separated by SDS-PAGE and then blotted with an anti-HA antibody and an anti-SPRY2 antibody.

## Tumorspheres assay

The tumorspheres assay was performed according to the vendor's (Stemcell Technologies, Vancouver, Canada) protocol. Briefly, single-cell suspensions were seeded in the six-well ultra-low attachment plate (Corning, New York, NY, USA) at a density of $3.5–4.0 \times 10^4$ cells in 2 mL of freshly prepared Complete MammoCult Medium (Stemcell Technologies) per well. After incubation for seven days, tumorspheres with diameter greater than 40 mm were counted.

## Statistical analysis

Unless otherwise noted, each sample was assayed in triplicate. Cell viability and migration/invasion assays were repeated three to four times. The in vitro biochemical and molecular biological experiments were repeated three times. Unless otherwise noted, data were presented as the mean ± s.e.m., and the two-tailed Student's t test was used to compare two groups. The differences were considered statistically significant when the P values were < 0.05.

## Results

It has been confirmed that miR-22 plays an anti-tumor role in liver cancer. Researchers have illuminated that miR-22 functions in HCC development through its regulation of different genes. In addition to being a direct target of miR-22, CBL is also an E3 ligase for SPRY2, an essential modulator of the ERK signaling pathway. We hypothesize that miR-22 may regulate SPRY2 by inhibiting CBL to establish its functional role in HCC development. To explore the existence of the miR-22/CBL/SPRY2 axis and the significance of this axis for the miR-22 function in HCC cell behavior, the effect of miR-22-3p on HCC EMT, migration/invasion, and CSC features was first studied. Immunoblot and ubiquitination assays demonstrated conclusively that miR-22-3p regulates SPRY2 at the protein level. The knockdown of SPRY2 in cells with overexpression of miR-22-3p eliminated the inhibitory effects of miR-22-3p on cell migration and invasion. Furthermore, miR-22-3p affects the ERK signaling pathway through its indirect regulation of SPRY2.

### miR-22-3p inhibits EMT, cell migration and invasion, and CSC features of liver cancer cells

It has been reported that miR-22-3p is downregulated in HCC [37] and is a suppressor of tumor growth and metastasis of HCC [37–39]. We question whether overexpression of miR-22-3p will affect the in vitro behavior of HCC cells. We first overexpressed miR-22-3p in HepG2 and HuH-7 HCC cells (**Fig 1A**) and then checked the expression of EMT marker genes and the migration and invasion capability of those cells. Cell viability assay indicated that there is no difference between miR-22-3p overexpressed cells and control cells (**Fig 1B**). In both HepG2 and HuH-7 cells, miR-22-3p caused upregulation of E-cadherin, the epithelial cell marker, and downregulation of N-cadherin, the typical mesenchymal cell marker (**Fig 1C**). These results indicated that miR-22-3p can inhibit the EMT process. Cell migration and invasion assays also showed that the capability of cell migration and invasion was decreased with the elevated miR-22-3p level (**Fig 1D and 1E**). To confirm the inhibitory effect of miR-22-3p on HCC cell migration and invasion, we knocked down miR-22-3p in HepG2 cells (**Fig 1F**). We found that there was an increase in cell migration and invasion capability (**Fig 1G**) after miR-22 was knocked down.

Cells going through EMT usually show the properties of CSC [45]. The induction of the EMT program can generate stem-like cells. We have shown that miR-22-3p can inhibit the EMT process. Is it possible that inhibiting miR-22-3p on EMT will alter the CSC characteristics of HCC cells? EpCAM is a well-known stem cell marker in HCC [46]. Consequently, we checked the expression of EpCAM in cells that overexpressed miR-22-3p using Western blotting. We found that miR-22-3p indeed decreased the expression of EpCAM in both HepG2 and HuH7 cells (**Fig 1H**). In accordance with the results of EpCAM expression, we also found that miR-22-3p reduced the tumorsphere-forming ability of both HepG2 and Huh7 cells (**Fig 1I**). The above results clearly demonstrated that miR-22-3p inhibited EMT, cell migration and invasion, and CSC features of liver cancer cells.

### miR-22-3p indirectly upregulates SPRY2 by inhibiting the expression of CBL

Since CBL is both a direct target of miR-22-3p and an E3 ligase of SPRY2, we are curious as to whether miR-22-3p can indirectly regulate SPRY2 by inhibiting CBL. To test our hypothesis, we examined the expression of SPRY2 by qPCR and Western blot in HePG2 cells expressing elevated levels of miR-22-3p. We did not find any difference in SPRY2 expression at the

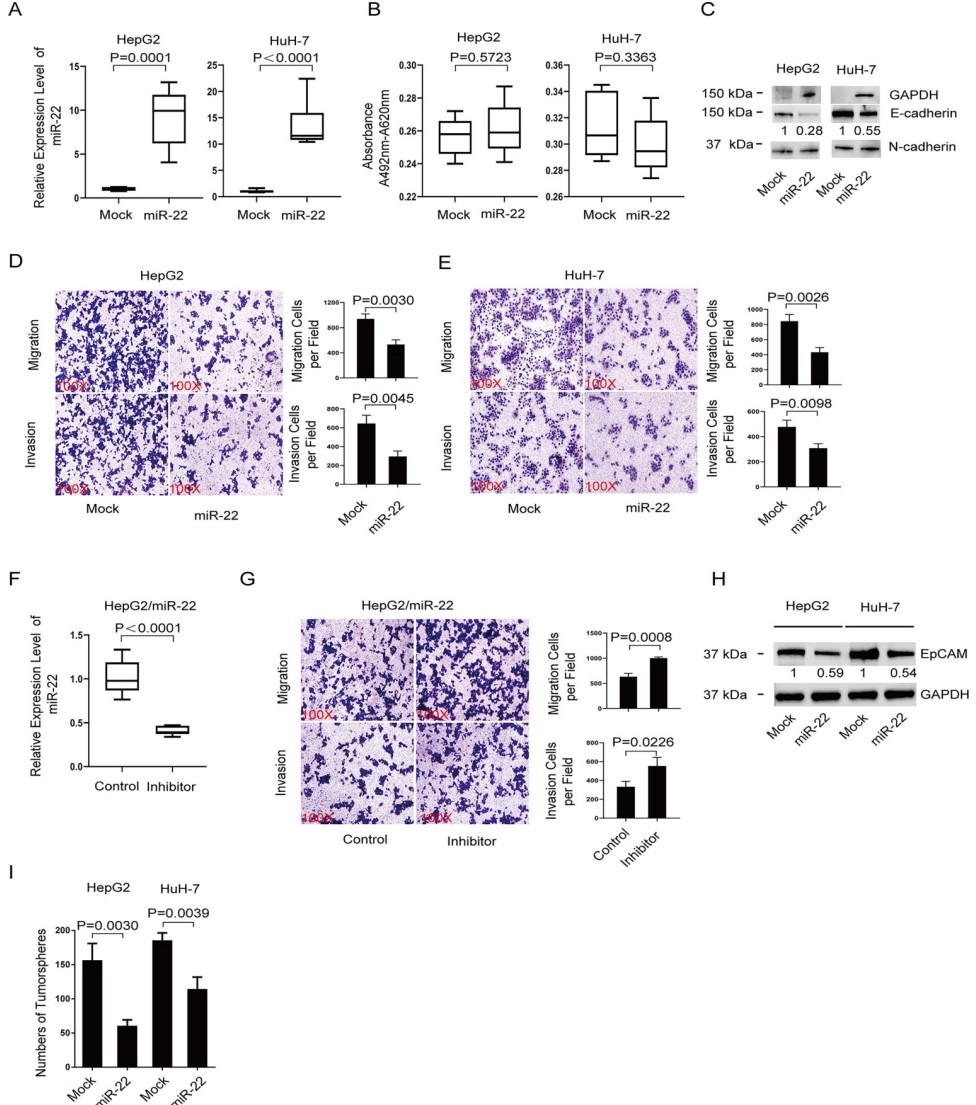

**Fig 1. miR-22-3p inhibits EMT, cell migration and invasion, and CSC features of HCC cells. A.** Expression level of miR-22-3p in the four cell lines: HepG2 mock cells, HepG2 overexpressing miR-22-3p cells, HuH-7 mock cells, and HuH-7 overexpressing miR-22-3p cells. **B.** Cell viability of the above four cell lines. Immunoblotting of E-cadherin and N-cadherin (**C**) and EpCAM (**H**) in the above four cell lines. **D and E.** Representative images and statistical results of Transwell migration assays and Matrigel invasion assays in the above four cell lines. **F.** Expression level of miR-22-3p in miR-22-3p-knocked-down cells and control cells. **G.** Representative images and statistical results of Transwell migration assays and Matrigel invasion assays of miR-22-3p knock-down cells and control cells. **I.** Quantification of tumorsphere formation by the above four cell lines. The data are presented as means ± SD of three independent experiments, and statistical significance was determined by two-tailed, unpaired Student's t test. The quantification result shown on each graph represents the statistical analysis of three experiments.

mRNA level (**Fig 2A**) between miR-22-3p overexpressing cells and control cells. However, we did observe that the expression of SPRY2 was indeed increased at the protein level (**Fig 2B**). When miR-22-3p was knocked down, SPRY2 expression was subsequently decreased (**Fig 2B**). To further verify our hypothesis, we performed the ubiquitination assay using miR-22-3p overexpressing cells and control cells. We found that the ubiquitination level of SPRY2 in miR-22-3p overexpressing cells is decreased (**Fig 2C**). These results strongly indicated that miR-22-3p can regulate SPRY2 by attenuating its ubiquitination. Furthermore, we performed

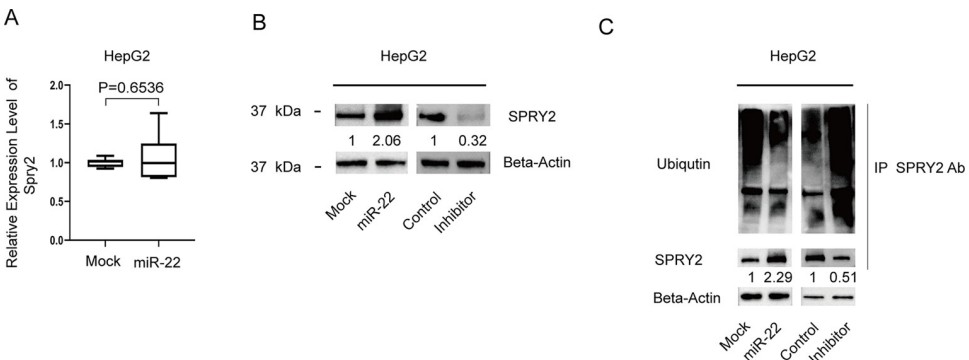

**Fig 2. miR-22-3p indirectly upregulates SPRY2 by inhibiting the expression of CBL. A.** qPCR of SPRY2 in HepG2 mock cells and HepG2 overexpressing miR-22-3p cells. **B.** Immunoblotting of SPRY2 in the four cell lines: HepG2 mock cells, HepG2 overexpressing miR-22-3p cells, HepG2 control cells, and miR-22-3p knockdown HepG2 cells. **C.** Ubiquitination level of SPRY2 in the above four cell lines.

a similar ubiquitination assay in miR-22-3p knockdown cells. Consistent with the findings in cells with overexpression of miR-22-3p, knockdown of miR-22-3p increased SPRY2 ubiquitination (**Fig 2C**).

## miR-22-3p/CBL/SPRY2 axis regulates EMT, migration, and invasion of liver cancer cells

To confirm the existence of the miR-22-3p/CBL/SPRY2 axis, we re-expressed CBL in cells with overexpression of miR-22-3p (**Fig 3A**, top panel). Western blot results indicated that the upregulation of SPRY2 by miR-22-3p can be reversed by re-expressing CBL in these cells (**Fig 3A**, middle panel). To determine whether the effects of miR-22-3p on EMT, cell migration, and invasion were mediated by SPRY2, we attempted to inhibit SPRY2 in miR-22-3p-overexpressing cells. We first checked the efficiency of SPRY2 knockdown plasmids and found that clone A is the most effective one (**Fig 3B**). This clone was subsequently transfected into miR-22-3p overexpressing cells to make a miR22/SPRY2 KD cell line (**Fig 3C**). The expression of EMT marker genes in these cells was evaluated by immunoblotting. We found that the knockdown of SPRY2 rescued the miR-22-3p overexpressed phenotype and led to the decrease of E-cadherins and the increase of N-cadherins (**Fig 3D**).

The aforementioned cells were also evaluated for their capacity for migration and invasion. The results showed that the inhibitory effect of miR-22-3p on cell migration and invasion can be reversed by knocking down SPRY2 (**Fig 3E**, right panel). miR-22-3p can regulate EMT, migration, and invasion of HCC cells through the miR-22-3p-CBL-SPRY2 axis, as demonstrated by the aforementioned findings.

## miR-22-3p suppresses ERK signaling by upregulating SPRY2

SPRY2 has been reported to be an inhibitor of MAPK/ERK signaling, which is important for tumor growth and metastasis [47]. After confirming that miR-22-3p could promote SPRY2 expression, we sought to determine whether miR-22-3p can affect MAPK/ERK signaling. We examined the expression of ERK and p-ERK in miR-22-3p-overexpression cells using immunoblotting. As shown in **Fig 4A**, overexpression of miR-22-3p decreased ERK protein phosphorylation. However, the phosphorylation levels of the ERK protein were increased in the miR-22-3p-knockdown cells (**Fig 4B**). While we knocked down SPRY2 in miR-22-3p-overexpressed cells, the phosphorylation levels of ERK proteins recovered to the basal level (**Fig 4C**).

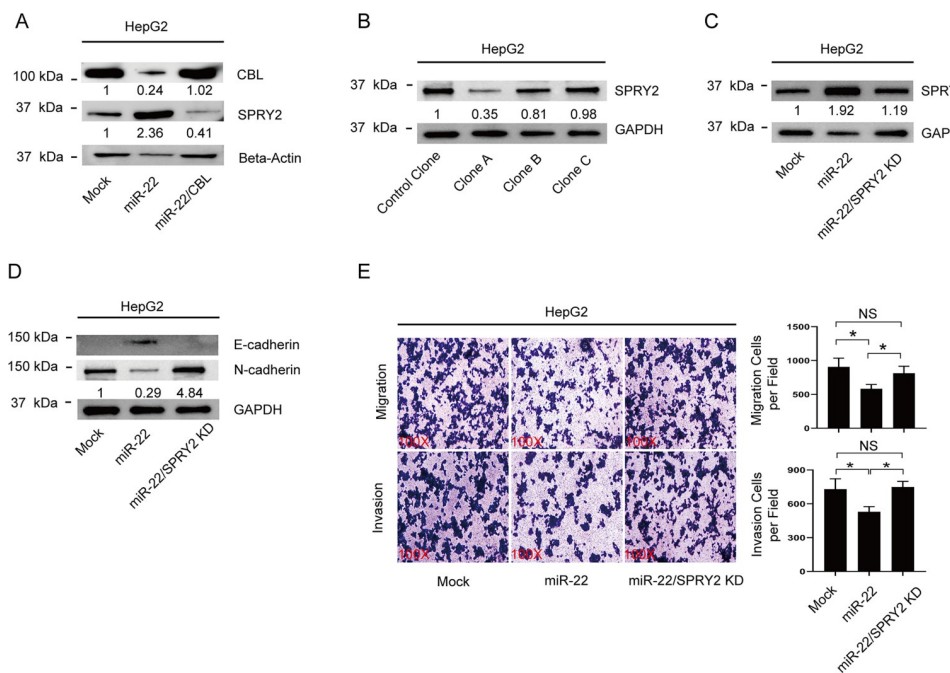

**Fig 3. miR-22-3p regulates HCC cell EMT, migration, and invasion through the miR-22-3p-CBL-SPRY2 axis. A.** Immunoblotting of CBL and SPRY2 in the three cell lines: HepG2 mock cells, HepG2 overexpressing miR-22-3p cells, and HepG2 overexpressing miR-22-3p plus CBL cells. **B.** Immunoblotting of SPRY2 in HepG2 overexpressing miR-22 cells that were transfected with different SPRY2 knockdown plasmids. Immunoblotting of SPRY2 (**C**) and E-cadherin and N-cadherin (**D**) in the three cell lines: HepG2 mock cells, HepG2 overexpressing miR-22-3p cells, and HepG2 overexpressing miR-22-3p/ SPRY2 knockdown cells. **E.** Representative images and statistical results of Transwell migration assays and Matrigel invasion assays of HepG2 mock cells, HepG2 overexpressing miR-22-3p cells, and HepG2 overexpressing miR-22-3p/ SPRY2 knockdown cells. The data are presented as means ± SD of three independent experiments, and statistical significance was determined by two-tailed, unpaired Student's t test. The quantification result shown on each graph represents the statistical analysis of the three experiments.

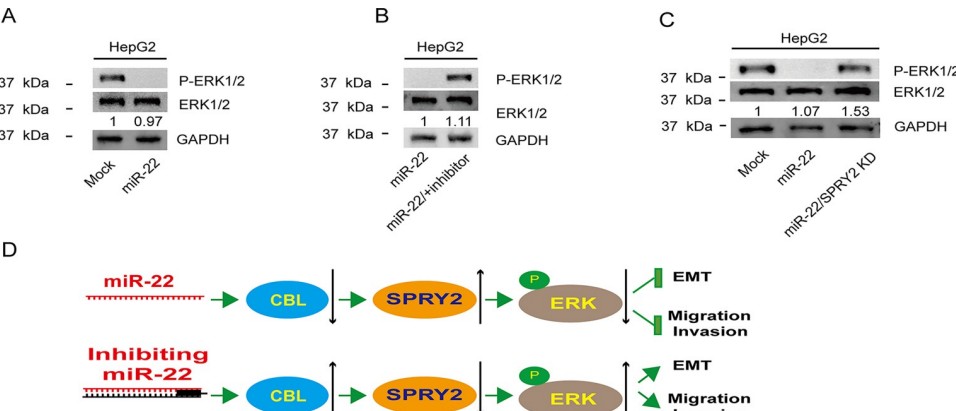

**Fig 4. miR-22-3p suppresses ERK signaling by upregulating the expression of SPRY2. A and B.** Immunoblotting of ERK and p-ERK in HepG2 mock cells, HepG2 overexpressing miR-22-3p cells, and HepG2 knockdown miR-22-3p cells. **C.** Immunoblotting of ERK and p-ERK in HepG2 mock cells, HepG2 overexpressing miR-22-3p cells, and HepG2 overexpressing miR-22-3p/ SPRY2 knockdown cells. **D.** miR-22-3p/CBL/SPRY2/ERK axis regulates EMT, migration and invasion of liver cancer cells.

Therefore, it is believed that miR-22-3p inhibits ERK signaling activity through indirect regulation of SPRY2 (Fig 4. miR-22-3p suppresses ERK signaling by upregulating the expression of SPRY2).

## Discussion

As important regulators of gene expression, microRNAs are involved in numerous physiological and pathological processes. Frequently, miRNA dysregulation leads to aberrant gene expression, which is associated with human diseases, including cancer. It has been recognized that governing the miRNA level may be a novel potential therapeutic strategy for diseases. Currently, two major approaches are used to develop miRNA-based disease therapies. One is to enhance the function of endogenous miRNAs using the miRNA mimics, and another is to inhibit harmful miRNA expression using small-molecule inhibitors, antagomiRs, and miRNA sponges. For example, restoring miR-33 and miR-145 was shown to suppress colon cancer development [48]. Silencing miR-296 prevents breast cancer development [49] and suppresses colorectal cancer EMT and metastasis [50].

miR-22 is an evolutionarily conserved miRNA that express in many tissues and organs and involves in many physiological and pathological processes. For example, it has been found that miR-22 involves in liver fibrosis [51], cardiac remodeling [52], development of autoimmune diseases [53], multiple sclerosis (MS) pathogenesis [54]. In terms of its function in cancer, miR-22 acts as either a tumor repressor or a tumor promoter depending on their tissue-specific expression patterns and their mRNA targets. One very interesting function for miR-22 is that it can exhibit the total opposite effects (tumor suppressor or tumor promoter) in different cancers by targeting the very same gene. For example, by targeting PTEN, miR-22 activates PI3K/AKT signaling pathway, resulting in downregulation of p27(-Kip1) and overexpression of Survivin and Ki-67 proteins in chronic lymphocytic leukemia B cells and acts as oncogene [55]. Similarly, miR-22 promotes prostate tumorigenesis through its inhibition on PTEN [56]. However, in clear cell renal cell carcinoma, miR-22 functions as an anti-tumor gene by suppressing PTEN [57]. What causes this interesting phenomenon is unknown. One simple explanation for this is that miR-22 may act on multiple targets in cells. Besides targeting PTEN, miR-22 also targets other genes. Those genes are different in different cancers. The overall effects of miR-22 on cancer are determined by the coordination of those genes. Although miR-22 targets PTEN in two different cancers, the other genes regulated by it in the two cancers are different, which may make miR-22 exhibit the opposite effects in two cancers by targeting the very same gene (PTEN).

So far, the low expression of miR-22 has been found in several cancers, such as bladder cancer [58], breast cancer [59–62], colon and colorectal cancer [63,64], gastric cancer [65,66], and liver cancer [24–26,67]. In most cancers, miR-22 exerts its tumor suppressing effect by suppressing its direct target genes. For example, Pan et al. recently reported that miR-22-3p acted as an anti-tumor agent in HCC by downregulating ETS1 [68]. miR-22 has two mature forms. miR-22-5p is a passenger strand and usually is degraded after being processed. Although it is a functional molecule, there is no evidence to link it to liver cancer. In addition, no experimental data indicates that miR-22-5p can target CBL. On the other hand, miR-22-3p is a functional guide strand when it is processed. A lot of studies have proved that it associates with liver cancer development. Therefore, we focused our study on miR-22-3p, not miR-22-5p. We found that miR-22 indirectly upregulates the expression of SPRY2 by inhibiting the expression of its direct target gene CBL, which encodes an E3 ligase to promote SPRY2 degradation. SPRY2 is a modulator of MAPK/ERK signaling which ubiquitously exists in eukaryotic cells and plays important roles in various physiological and pathological processes. SPRY2 functions as a

tumor suppressor in most cancers and its expression has been found to be low in a variety of cancers, such as prostate cancer [69], breast cancer [70], lung cancer [71], liver cancer [33] and colon cancer [72]. SPRY2 can inhibit epithelial tumor cell EMT [73] as well as cell migration and invasion [74]. Our results confirmed that SPRY2 is indirectly regulated by miR-22-3p in HCC cells, and through the miR-22-3p/CBL/SPRY2 axis, miR-22-3p suppresses EMT, migration, and invasion of HCC cells. The rescue experiments demonstrated that either re-expression of CBL or knockdown of SPRY2 in cells with overexpression of miR-22-3p reversed the inhibitory effect of miR-22-3p on EMT and cell migration and invasion. Compared to control cells, the ubiquitination level of SPRY2 was lower in miR-22-3p cells, indicating that miR-22-3p can regulate SPRY2 post-translational modification. Indeed, SPRY2 is more stable in miR-22-3p cells.

Numerous studies had confirmed that SPRY2 primarily served its biological function by inhibiting canonical MAPK/ERK signaling [30]. In our current study, we found that miR-22-3p exerted its ERK inhibitory function by indirectly upregulating SPRY2. miR-22-3p overexpression was associated with low p-ERK expression, whereas miR-22-3p inhibition was associated with high p-ERK expression. In addition to validating the inhibitory role of miR-22-3p for EMT, cell migration, and invasion in HCC cells, our study uncovered a novel molecular mechanism by which miR-22-3p contributes to HCC development.

miRNA-based therapeutics represents an attractive approach for the treatment of cancers, as well as many other diseases. However, before this treatment can be translated to the clinic, we should fully understand the molecular mechanism of the specific miRNA. Our current studies expand our understanding on how miR-22 may exert its tumor suppression. Many studies have indicated that miR-22 may serve as a promising therapeutic target for precision treatments in different cancers. The use of miR-22 as a therapeutic target has two advantages. First, miR-22 is naturally occurring molecule in human cells, and therefore it can be properly processed. Second, miR-22 acts by targeting multiple genes within one pathway, thus causing a broader yet specific response. However, before the translation of miR-22 based therapeutics into the clinic, some issues have to be solved. These issues include delivery (molecule stability and efficient intracellular delivery), specificity (on-target effects or off-target effects) and tolerability. With the technical advancements in the related field, we believe that all the challenges will finally be overcome.

## Conclusion

In conclusion, by directly repressing CBL, miR-22-3p decreases the ubiquitination of SPRY2 and stabilizes it, which inhibits the EMT, migration/invasion, and CSC characteristics of HCC cells. This study elucidated a novel regulatory mechanism of miR-22 in HCC development. miR-22 might be a promising therapeutic target for HCC treatment.

## Supporting information

**S1 Raw images. All original images for blots and gels.**
(PDF)

## Author Contributions

**Investigation:** Shuaishuai Cui, Yuanyuan Chen, Yunfei Guo, Xing Wang, Dahu Chen.

**Methodology:** Shuaishuai Cui, Yuanyuan Chen, Yunfei Guo, Xing Wang, Dahu Chen.

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
