## [Decision Letter · Decision Letter 0]

13 Sep 2022

PONE-D-22-16558

Hsa-miR-22-3p inhibits liver cancer cell EMT and cell migration/ invasion by indirectly regulating SPRY2

PLOS ONE

Dear Dr. Chen,

Thank you for submitting your manuscript to PLOS ONE. After careful consideration, we feel that it has merit but does not fully meet PLOS ONE’s publication criteria as it currently stands. Therefore, we invite you to submit a revised version of the manuscript that addresses the points raised during the review process.

Please, follow the suggestions form the reviewers to improve the manuscript. In particular, include a rationale to studying miR-22-3 in this context. Also, add more detailed information to the materials and methods: One of the PLOS ONE publication criteria is that “Experiments, statistics, and other analyses are performed to a high technical standard and are described in sufficient detail.” This implies that other researcher can reproduce the experiments described. In its current form, this is not the case as pointed out by reviewer 1. In addition, you should correct the figures and change from bar or column graph to box or dots plots, so the readers can identify the data points.

I also request that you proof-read your manuscript  and expand the discussion, especially as is difficult to evaluate whether the described "axis" is relevant in vivo and what the normal role of miR-22 in the liver might be. A context and possibly explaination should be added to why other studies using KO cells (e.g., PMID: 25323629) show more or less exactly the opposite. It is essential to include how a microRNA such as miR-22 can have fundamentally different effects on the very same genes in two different settings. 

We look forward to receiving your revised manuscript.

Kind regards,

Claudia D. Andl, Ph.D.

Academic Editor

PLOS ONE

Journal Requirements:

2. Please comment on the relationship between this manuscript and the published article available at: https://portlandpress.com/bioscirep/article/41/8/BSR20210318/229438/miR-219-5p-targets-TBXT-and-inhibits-breast-cancer. Please comment on the degree of redundancy with your related article, and about the contribution of the PLOS ONE submission to the base of scientific knowledge in light of the research presented in the related article.

 "This study was supported by National Natural Science Foundation of China (81872005) to DC."  

5. PLOS ONE now requires that authors provide the original uncropped and unadjusted images underlying all blot or gel results reported in a submission’s figures or Supporting Information files. This policy and the journal’s other requirements for blot/gel reporting and figure preparation are described in detail at https://journals.plos.org/plosone/s/figures#loc-blot-and-gel-reporting-requirements and https://journals.plos.org/plosone/s/figures#loc-preparing-figures-from-image-files. When you submit your revised manuscript, please ensure that your figures adhere fully to these guidelines and provide the original underlying images for all blot or gel data reported in your submission. See the following link for instructions on providing the original image data: https://journals.plos.org/plosone/s/figures#loc-original-images-for-blots-and-gels. In your cover letter, please note whether your blot/gel image data are in Supporting Information or posted at a public data repository, provide the repository URL if relevant, and provide specific details as to which raw blot/gel images, if any, are not available. Email us at plosone@plos.org if you have any questions."

Reviewers' comments:

Reviewer's Responses to Questions

**Comments to the Author**

1. Is the manuscript technically sound, and do the data support the conclusions?

Reviewer #1: Yes

Reviewer #2: Yes

2. Has the statistical analysis been performed appropriately and rigorously? 

Reviewer #1: No

Reviewer #2: Yes

3. Have the authors made all data underlying the findings in their manuscript fully available?

Reviewer #1: Yes

Reviewer #2: Yes

4. Is the manuscript presented in an intelligible fashion and written in standard English?

Reviewer #1: Yes

Reviewer #2: Yes

5. Review Comments to the Author

Reviewer #1: Summary: This paper attempts to elucidate the mechanism and function of miR-22 in liver cancer. While this miRNA has different known functions depending on the cancer, several studies have shown that in hepatocellular carcinoma (HCC) there is a marked decrease in miR-22, which suggests that miR-22 can function as a tumor suppressor. In this paper, the authors link miR-21-3p with direct inhibition of CBL, which stabilizes SPRY2 and prevents the epithelial mesenchymal transition (EMT) in HCC cells. Please address the points below.

Major Points:

• On page 4, in the introduction, there is no differentiation between miR-22-5p versus miR-22-3p. This is a key element that should be explained in the introduction, especially since the 3p and 5p functions of miRNAs are still being examined. The rationale is also missing as to how they choose miR-22-3p as their target of interest.

• Make sure that the verb tense is consistent throughout the paper. Most of the paper is in past tense, but there are present tense verbs present in several pages (4 and 12).

• In page 4, the authors mention senescence-like phenotypes in HCC cells, but how does that connect to the anti-tumor role miR-22 has in HCC?

• In the introduction on page 5, the paper discusses the MAPK signaling pathway briefly, but it should be made clear what the connection between ERK and the MAPK signaling pathway are. Also, how is SPRY2 a modulator of this pathway? At what step? Is CBL also part of the MAPK signaling pathway?

• For the migration and invasion assays, the quantification was done manually by counting cells in the images. Is there an alternative method of quantification such as ImageJ or a cell counting software that could be more objective?

• In figure 1D, 1E, and 1G, the y-axes are missing units and titles.

• The migration and invasion images in figures 1 and 3 are all missing scale bars. Also, arrows may be useful for pointing out significant movement or lack thereof of the cells.

• In figure 1, how were the cells quantified? Were they counted as groups or individual cells per frame?

• In figure 1, why is Huh7 not included in the cell migration and invasion assays? Did they knockdown and/or overexpress miR-22-3p in these cells as well?

• On page 16, it is not clear how CBL was re-expressed in miR-22-3p overexpression cells was done? Lenti-viral vector? Plasmid?

• In figure 4, there is no statistical analyses for the western blot.

• For figure 4D, the authors could also show, visually, how knocking out or inhibiting miR-22 would have the opposite effect.

• Limitations and further applications need to be discussed in the conclusion.

• They mention that miR-22 can be a potential biomarker, but are there other applications? Clinical or otherwise?

Minor Points:

• In the beginning of the abstract, there could be more of a brief explanation of how miR-22 can function as a tumor suppressor or promoter, like through what mechanisms?

• Several times in the paper, the “r” in miR-22 is lowercase when it should be uppercase. Or the “m” is uppercase when it should be lowercase. This is notable in pages 4, 5, and 12.

• At the top of page 4, it should be “miR-22 is one such dual-functioning miRNA in cancer”.

• A brief definition of ubiquination somewhere in the introduction would be helpful for the readers.

• In the materials and methods section, there is no catalog number for DMEM (high glucose).

• The cells are not cited properly in the cell culture section of the materials and methods.

• On page 11, “To explore the existence of the miR-22/CBL/SPRY2 axis and the significance of this axis for the miR-22 function in HCC cell behavior, the effect of miR-22-3p on HCC EMT, migration/invasion, and CSC features were first studied.”

• On page 15, qPCR not QPCR.

• In figure 3, some of the graphs do not have standard deviations shown or p-values (3A, B, C, D).

• For the Western blot, the abbreviation for kilodalton is kDa, not kd.

Reviewer #2: The manuscript describes the impact of a well studied microRNA, miR-22, on the biology of two liver cancer cel lines. Although well established that this microRNA has inhibitory effects on liver cancer cells, the authors add some interesting data on how miR-22 may exert its tumor suppression.

The data appear robust, the line of thought is clear, and the presentation of the data convincing.

Although not much new is added to our understanding of microRNA biology and HCC, this may be true to many similar manuscripts. It is difficult to evaluate whether the described "axis" is relevant in vivo and what the normal role of miR-22 in the liver might be. Furthermore, cleaner experiments in other tumor types using KO cells (e.g., PMID: 25323629) show more or less exactly the opposite.

It would be nice to hear in the discussion how a microRNA such as miR-22 can have fundamentally different effects on the very same genes in two different settings. Overall, a more critical discussion may be helpful.

Furthermore, there are too many bar graphs. bar graphs hide data and therefore box plots are preferable and requested.

In summary: better plots, better discussion.

6. PLOS authors have the option to publish the peer review history of their article (what does this mean?). If published, this will include your full peer review and any attached files.

Reviewer #1: No

Reviewer #2: No

---

## [Author Response · Author response to Decision Letter 0]

14 Nov 2022

Reviewer 1:

Major Points:

• On page 4, in the introduction, there is no differentiation between miR-22-5p versus miR-22-3p. This is a key element that should be explained in the introduction, especially since the 3p and 5p functions of miRNAs are still being examined. The rationale is also missing as to how they choose miR-22-3p as their target of interest.

RE: To address the reviewer’s concerns, we provided the rationale for our focusing on miR-22-3p, not miR-22-5p, in liver cancer and re-wrote the introduction part. The changes have been highlighted in the new manuscript. Please see it in Page 4-5, Line 59-74; Page 7, Line 124-129 and Page 21-22, Line 411-416. 

One more thing we need to pay attention to is that in miRbase database, miR-22-5p ID is miR-22*. miR-22-3p ID is miR-22. Therefore, when a research paper mentions miR-22 (not specifically indicates miR-22-5p or miR-22-3p), we think the target studied should be miR-22-3p.

• Make sure that the verb tense is consistent throughout the paper. Most of the paper is in past tense, but there are present tense verbs present in several pages (4 and 12).

RE: We have revised and unified the verb tense in the paper.

• In page 4, the authors mention senescence-like phenotypes in HCC cells, but how does that connect to the anti-tumor role miR-22 has in HCC?

RE: Cellular senescence leads to a senescent phenotype, which is characterized by upregulated senescence-associated β-galactosidase (SA-β-gal) activity, cell cycle arrest and cell proliferation inhibition. It is reported that cellular senescence is a key obstacle to the initiation and progression of HCC. Sirt1, a direct target of miR-22-3p, has anti-senescence effect. miR-22-3p plays an anti-tumor role in liver cancer by inhibiting the expression of Sirt1, which decrease the anti-senescence capability of Sirt1. (Zhao etc. lncRNA miat functions as a ceRNA to upregulate sirt1 by sponging miR-22-3p in HCC cellular senescence. doi: 10.18632/aging.102240.)

• In the introduction on page 5, the paper discusses the MAPK signaling pathway briefly, but it should be made clear what the connection between ERK and the MAPK signaling pathway are. Also, how is SPRY2 a modulator of this pathway? At what step? Is CBL also part of the MAPK signaling pathway?

RE: We have addressed the reviewer’s questions in introduction part. Please see it in Page 5-6, Line 87-99 and Line 105-107; Page 7, Line 124-129. 

CBL is not a part of MAPK signaling but it can regulate this signaling pathway by modified the upstream signal molecule, such as EGFR, etc..

• For the migration and invasion assays, the quantification was done manually by counting cells in the images. Is there an alternative method of quantification such as ImageJ or a cell counting software that could be more objective?

RE: We have not found a better way to do this. Counting cells in the images is a generally accepted method for this kind of analysis. We have published similar results in Nature Medicine, PLOS genetics and some other journals.

• In figure 1D, 1E, and 1G, the y-axes are missing units and titles.

RE: Following the reviewer’s instruction, we have added the missing Y-axes in Figure 1.

• The migration and invasion images in figures 1 and 3 are all missing scale bars. Also, arrows may be useful for pointing out significant movement or lack thereof of the cells.

RE: Because of the limited condition and the outdated microscope we used, the images took from our old microscope do not show the scale bar automatically, therefore we have added the magnification in each image. We hope this will meet the reviewer’s requirement. 

 We are not sure where we should add arrows in each image. We checked some papers that reported the similar migration and invasion results and did not find the useful information. We would appreciate it if the reviewer could provide detailed instruction for this.

• In figure 1, how were the cells quantified? Were they counted as groups or individual cells per frame?

RE: The cells were counted as individual cells per frame. 

• In figure 1, why is Huh7 not included in the cell migration and invasion assays? Did they knockdown and/or overexpress miR-22-3p in these cells as well?

RE: We did include HuH7 cell migration and invasion results in Figure1. Please see it in Figure 1E. 

 Figure 1E shows the migration and invasion behaviors of HuH7 cells when miR-22-3p is overexpressed. 

• On page 16, it is not clear how CBL was re-expressed in miR-22-3p overexpression cells was done? Lenti-viral vector? Plasmid?

RE：The CBL ORF sequence was cloned into the pBABE-puro vector, which is a retro-virus vector. 

• In figure 4, there is no statistical analyses for the western blot.

RE: We have added the quantitative analysis for some western blots.

• For figure 4D, the authors could also show, visually, how knocking out or inhibiting miR-22 would have the opposite effect.

RE: We have made a new Figure 4D following the reviewer’s instruction. 

• Limitations and further applications need to be discussed in the conclusion.

RE: We believe the further application for miR-22 is to use it as a therapeutic target in some cancers. We have briefly discussed this issue and pointed out the limitations or drawbacks regarding the clinical translation of this application. Please see it in Page 23, Line 441 to 454. 

• They mention that miR-22 can be a potential biomarker, but are there other applications? Clinical or otherwise?

RE: We addressed the questions of the reviewers here. In the meantime, some of the information and explanation have been included in discussion part in our revised manuscript.

Applications of miR-22:

miR-22 may well be promising as an independent early diagnostic biomarker for some cancers. For example, it has been found that the level of miR-22 in serum of hepatocellular cancer patients with hepatitis C virus is low (doi: 10.1007/s13277-016-5097-8, doi: 10.1038/sj.bjc.6605895.). On the contrary, the expression level of miR-22 in the serum of pancreatic patients is elevated (doi: 10.4251/wjgo.v6.i1.22.). 

In addition, the expression change of miR-22 in body fluids can be used to monitor the therapeutic effects. It has been found that the higher level of miR-22 in the serum of non-small cell lung cancer is associated with cancer aggression and the responsiveness status to the chemotherapeutic drug pemetrexed (doi: 10.4251/wjgo.v6.i1.22.)

One more potential application for miR-22 is that it can be used to increase chemoresenitiivty of anticancer drugs in cancer treatment. It has been found that miR-22 increases chemosensitivity to some anticancer drugs by directly targeting and activating or inactivating various downstream genes. For instance, miR-22 can enhance the chemoresensitivity of p53-mutant colon cancer cells to paclitaxel by regulating PTEN (doi: 10.1007/s11010-011-0872-8.).

Minor Points:

• In the beginning of the abstract, there could be more of a brief explanation of how miR-22 can function as a tumor suppressor or promoter, like through what mechanisms?

RE：To follow the suggestion of the reviewers, we have added a brief explanation in Abstract part. Please see Page2, Line 12 to 17. 

The general mechanism for microRNAs to play biological function is through their inhibition on the expression of their target genes. miR-22 functions as either a tumor promoter or a tumor repressor by inhibiting (directly or indirectly) the expression of anti-cancer genes or oncogenes, respectively. Due to its regulations on the important genes, miR-22 may accelerate cell senescence, block angiogenesis, decrease energy supplies, repress tumor cell cycle and promote apoptosis. On the other hand, miR-22 can modulate tumor suppressor molecules to activate oncogene relevant signaling pathway. By this way, miR-22 can play either a tumor repressor role or a tumor promoter role depending on its tissue-specific expression patterns and its mRNA targets. 

• Several times in the paper, the “r” in miR-22 is lowercase when it should be uppercase. Or the “m” is uppercase when it should be lowercase. This is notable in pages 4, 5, and 12.

RE: We have corrected these mistakes. 

• At the top of page 4, it should be “miR-22 is one such dual-functioning miRNA in cancer”.

RE: We have deleted this sentence. 

• A brief definition of ubiquination somewhere in the introduction would be helpful for the readers.

RE: We have added some brief introduction for ubiquination in the introduction part. Please see it in Page 7, Line 116 to 122.

• In the materials and methods section, there is no catalog number for DMEM (high glucose).

RE: We have added the catalog number for DMEM (high glucose), Please see it in Page 8, Line 142. 

• The cells are not cited properly in the cell culture section of the materials and methods.

RE: We have provided the related information for those cell lines. 

• On page 11, “To explore the existence of the miR-22/CBL/SPRY2 axis and the significance of this axis for the miR-22 function in HCC cell behavior, the effect of miR-22-3p on HCC EMT, migration/invasion, and CSC features were first studied.”

RE: We have corrected this sentence. Please see it in Page 13, 241-244.

• On page 15, qPCR not QPCR.

RE: We have changed it. 

• In figure 3, some of the graphs do not have standard deviations shown or p-values (3A, B, C, D).

RE: We have added p-values for some graphs. 

• For the Western blot, the abbreviation for kilodalton is kDa, not kd.

RE: We have changed kd to kDa for all western blots to follow the instruction of the reviewer. 

Reviewer #2: The manuscript describes the impact of a well studied microRNA, miR-22, on the biology of two liver cancer cell lines. Although well established that this microRNA has inhibitory effects on liver cancer cells, the authors add some interesting data on how miR-22 may exert its tumor suppression.

The data appear robust, the line of thought is clear, and the presentation of the data convincing.

RE: We are encouraged that the reviewers found that “the authors add some interesting data on how…. and the data appear robust, the line of thought is clear, and the presentation of the data convincing.” 

Although not much new is added to our understanding of microRNA biology and HCC, this may be true to many similar manuscripts. It is difficult to evaluate whether the described "axis" is relevant in vivo and what the normal role of miR-22 in the liver might be. Furthermore, cleaner experiments in other tumor types using KO cells (e.g., PMID: 25323629) show more or less exactly the opposite.

RE: We do agree with the reviewer’s points. Indeed, we just showed that there is a miR-22-3p/CBL/Spry2 axis in vitro, but not in vivo. More studies are needed to prove if there is such a pathway in vivo. This is one part of our ongoing project. We hope to get some solid data and produce a new paper in near future to address the concerns of the reviewer. 

Tang’s (PMID: 25323629) work about the role of miR-22 in gastric cancer is excellent. They reported that miR-22 can repress gastric cancer metastasis by targeting MTDH and indicated that miR-22 may be a tumor suppressor in gastric cancer. The role of miR-22 in gastric cancer they reported is similar to the role of miR-22-3p in liver cancer we presented in our manuscript. 

By carefully reading the paper (PMID: 25323629. microRNA-22 acts as a metastasis suppressor by targeting metadherin in gastric cancer. Yunyun Tang Xiaoping Liu, Bo Su, Zhiwei Zhang, Xi Zeng, Yanping Lei, Jian Shan, Yongjun Wu, Hailin Tang, Qi Su), we did not find that the authors reported the opposite results that the reviewer mentioned in his/her comments. In addition, the authors did not use any KO cells in their work.

• It would be nice to hear in the discussion how a microRNA such as miR-22 can have fundamentally different effects on the very same genes in two different settings. Overall, a more critical discussion may be helpful.

RE: It is a very good suggestion. We have added the related information in the discussion part. Please see it in Page 20-21, Line 386-405. 

Furthermore, there are too many bar graphs. bar graphs hide data and therefore box plots are preferable and requested.

In summary: better plots, better discussion.

RE: Following the instruction of the reviewer, we have changed many bar graphs (for QPCR results）to box plots. As for the statistical results of cell migration/ invasion, and tumorsphere formation, we still use the bar graphs. After searching a lot of published papers, we find that the bar graphs are the prevailing way to show such results.

---

## [Decision Letter · Decision Letter 1]

26 Jan 2023

Hsa-miR-22-3p inhibits liver cancer cell EMT and cell migration/ invasion by indirectly regulating SPRY2

PONE-D-22-16558R1

Dear Dr. Chen,

We’re pleased to inform you that your manuscript has been judged scientifically suitable for publication and will be formally accepted for publication once it meets all outstanding technical requirements. Please note that I have acted as a reviewer for this manuscript, and you will find my comments below, under Reviewer 3.

Kind regards,

Atsushi Hosui, PhD, MD

Academic Editor

PLOS ONE

Additional Editor Comments (optional):

All criticisms and suggestions written by reviewers were responded and addressed. This manuscript is novel and worth reading.

Reviewers' comments:

Reviewer's Responses to Questions

**Comments to the Author**

1. If the authors have adequately addressed your comments raised in a previous round of review and you feel that this manuscript is now acceptable for publication, you may indicate that here to bypass the “Comments to the Author” section, enter your conflict of interest statement in the “Confidential to Editor” section, and submit your "Accept" recommendation.

Reviewer #1: All comments have been addressed

Reviewer #3: All comments have been addressed

2. Is the manuscript technically sound, and do the data support the conclusions?

Reviewer #1: Yes

Reviewer #3: Yes

3. Has the statistical analysis been performed appropriately and rigorously? 

Reviewer #1: N/A

Reviewer #3: Yes

4. Have the authors made all data underlying the findings in their manuscript fully available?

Reviewer #1: Yes

Reviewer #3: Yes

5. Is the manuscript presented in an intelligible fashion and written in standard English?

Reviewer #1: Yes

Reviewer #3: Yes

6. Review Comments to the Author

Reviewer #1: The authors have addressed all the questions sufficiently. I am satisfied with the responses and revision.

Reviewer #3: All comment by reviewers have been addressed. This manuscript has been improved and becomes worth reading for readers.

7. PLOS authors have the option to publish the peer review history of their article (what does this mean?). If published, this will include your full peer review and any attached files.

Reviewer #1: No

Reviewer #3: No

---

## [Editor Report · Acceptance letter]

30 Jan 2023

PONE-D-22-16558R1 

Hsa-miR-22-3p inhibits liver cancer cell EMT and cell migration/ invasion by indirectly regulating SPRY2 

Dear Dr. Chen:

I'm pleased to inform you that your manuscript has been deemed suitable for publication in PLOS ONE. Congratulations! Your manuscript is now with our production department. 

Kind regards, 

on behalf of

Dr. Atsushi Hosui 

Academic Editor

PLOS ONE